# Predictive Value of Fatty Liver Index for Long-Term Cardiovascular Events in Patients Receiving Liver Transplantation: The COLT Study

**DOI:** 10.3390/biomedicines11102866

**Published:** 2023-10-23

**Authors:** Alfredo Caturano, Gaetana Albanese, Anna Di Martino, Carmine Coppola, Vincenzo Russo, Raffaele Galiero, Luca Rinaldi, Marcellino Monda, Raffaele Marfella, Ferdinando Carlo Sasso, Teresa Salvatore

**Affiliations:** 1Department of Advanced Medical and Surgical Sciences, University of Campania Luigi Vanvitelli, 80138 Naples, Italyraffaele.marfella@unicampania.it (R.M.);; 2Department of Experimental Medicine, University of Campania Luigi Vanvitelli, 80138 Naples, Italy; 3Area Stabiese Hospital, 80053 Naples, Italy; 4Sbarro Institute for Cancer Research and Molecular Medicine, Center for Biotechnology, College of Science and Technology, Temple University, Philadelphia, PA 19122, USA; 5Division of Cardiology, Department of Medical Translational Sciences, University of Campania Luigi Vanvitelli, 80131 Naples, Italy; 6Department of Medicine and Health Sciences “Vincenzo Tiberio”, University of Molise, 86100 Campobasso, Italy; 7Department of Precision Medicine, University of Campania Luigi Vanvitelli, 80138 Naples, Italy

**Keywords:** fatty liver index, orthotopic liver transplantation, cardiovascular disease, myocardial infarction, stroke

## Abstract

Background and aims: Cardiovascular disease (CVD) is the leading cause of early mortality in orthotopic liver transplantation (OLT) patients. The fatty liver index (FLI) is strongly associated with carotid and coronary atherosclerosis, as well as cardiovascular mortality, surpassing traditional risk factors. Given the lack of data on FLI as a predictor of cardiovascular events in OLT recipients, we conducted a retrospective study to examine this topic. Methods and results: We performed a multicenter retrospective analysis of adult OLT recipients who had regular follow-up visits every three to six months (or more frequently if necessary) from January 1995 to December 2020. The minimum follow-up period was two years post-intervention. Anamnestic, clinical, anthropometric and laboratory data were collected, and FLI was calculated for all patients. Clinical trial.gov registration ID NCT05895669. A total of 110 eligible patients (median age 57 years [IQR: 50–62], 72.7% male) were followed for a median duration of 92.3 months (IQR: 45.7–172.4) post-liver transplantation. During this period, 16 patients (14.5%) experienced at least one adverse cardiovascular event (including fatal and non-fatal myocardial infarction and stroke). Receiver Operating Characteristic (ROC) analysis identified a cut-off value of 66.0725 for predicting cardiovascular events after OLT, with 86.7% sensitivity and 63.7% specificity (68% vs. 31%; *p* = 0.001). Kaplan–Meier analysis showed that patients with FLI > 66 had significantly reduced cardiovascular event-free survival than those with FLI ≤ 66 (log-rank: 0.0008). Furthermore, multivariable Cox regression analysis demonstrated that FLI > 66 and pre-OLT smoking were independently associated with increased cardiovascular risk. Conclusions: Our findings suggest that FLI > 66 and pre-OLT smoking predict cardiovascular risk in adult OLT recipients.

## 1. Introduction

Cardiovascular disease (CVD) represents a major concern for individuals who have undergone orthotopic liver transplantation (OLT), with statistics indicating that it is the leading cause of early mortality post-OLT, accounting for approximately 40% of such cases. Following CVD, infections and graft failure stand as the second and third most common causes of early mortality, constituting 28% and 12%, respectively [1]. Beyond the immediate post-transplant period, CVD, along with malignancies and end-stage kidney disease, continues to pose a significant threat, contributing significantly to long-term non-graft-related morbidity and mortality.

A decade ago, a comprehensive meta-analysis involving 12 studies shed light on the extent of this cardiovascular risk in OLT patients. The analysis revealed a substantial 10-year risk of 13.6% for cardiovascular events in these individuals [2]. Furthermore, a study conducted by Albeldawi et al. indicated cumulative risks of 4.5% and 10.1% for cardiovascular events within 1 and 3 years post-OLT, respectively [3]. Meanwhile, Fussner et al. found that over a longer time frame, 10.6%, 20.7%, and 30.3% of OLT recipients developed CVD within 1, 5, and 8 years after transplantation, respectively [4]. It is noteworthy that these frequencies are notably higher than what is observed in the general population, underscoring the unique cardiovascular challenges faced by OLT recipients.

Several factors are anticipated to further exacerbate the severity of cardiovascular risk in this population. One significant factor is the increasing age of OLT recipients, with a growing number of individuals aged over 65 undergoing transplantations. Moreover, improved OLT-related care has led to increased survival rates, resulting in a larger pool of long-term recipients at risk. The chronic use of immunosuppressive medications also contributes to heightened cardiovascular risk. Additionally, the rising prevalence of liver transplantation for nonalcoholic steatohepatitis (NASH) adds another layer of complexity to this issue [5,6,7,8].

Despite the growing awareness of the cardiovascular burden faced by post-OLT individuals, a consensus has yet to be reached regarding the optimal assessment of cardiovascular risk before intervention [9,10,11]. In particular, the distinct lack of a straightforward, accurate, and objective system for identifying patients at a higher risk of experiencing long-term cardiovascular events remains a notable research gap in the management of liver transplant recipients. The motivation lies in a number of factors, including the absence of consensus on outcome definition, incomplete knowledge, suboptimal data quality, and the uneven predictive accuracy of different non-invasive tests, as highlighted by a systematic review of the literature [12]. Preliminary findings from a recent study offer a glimmer of hope, suggesting that aortic pulse wave velocity, a surrogate marker of arterial stiffness, could serve as a valuable biomarker for assessing cardiovascular risk in OLT candidates [13]. In 2018, a predictive model known as the CAR-OLT score was developed to estimate the one-year global risk of death or hospitalization due to a significant cardiac or vascular event following OLT [14]. The CAR-OLT score is derived from easily accessible pre-transplant patients’ characteristics and personal history. It serves as a valuable tool for healthcare professionals to facilitate on-the-spot conversations regarding the one-year risk of significant cardiac or vascular events. However, it is important to note that CAR-OLT still requires external validation and does not address the long-term cardiac risk associated with OLT. Therefore, CAR-OLT should not be utilized for making decisions related to transplant management. Instead, it should be used to provide valuable information during risk-related discussions.

Nonalcoholic fatty liver disease (NAFLD), currently the most prevalent chronic liver disorder, has emerged as a risk factor in the general population for both severe hepatic consequences, such as cirrhosis and hepatocellular carcinoma, and cardiovascular morbidity and mortality. This risk extends to transplant recipients, particularly those with NASH [15,16,17,18,19].

While liver biopsy remains the gold standard for diagnosing NAFLD, efforts have been made to explore non-invasive and non-imaging approaches, which can be more practical in clinical settings. Among these methodologies, the fatty liver index (FLI) has gained substantial recognition as a simple yet accurate tool for identifying the presence of steatosis in both the general population and patients with NAFLD [20,21,22,23,24]. Importantly, FLI-assessed fatty liver has demonstrated a strong association with carotid and coronary atherosclerosis, as well as cardiovascular mortality, often surpassing the predictive power of traditional risk factors [25,26]. A recent study conducted in Korea, utilizing a large dataset, further confirmed the prognostic value of FLI in identifying in the general population those individuals at higher risk for cardiovascular events, including cardiovascular deaths [27].

Given the paucity of data regarding the significance of FLI, a well-validated surrogate marker of liver steatosis, as a predictor of cardiovascular events in OLT recipients, we have undertaken a comprehensive study to investigate this relevant topic. This research aims to shed light on the potential utility of FLI in assessing cardiovascular risk in the specific context of liver transplant recipients, with the ultimate goal of improving the management and outcomes of these individuals.

## 2. Patients and Methods

The “Cardiovascular outcomes in Orthotopic Liver Transplant patients study” (COLT study) is a multicenter retrospective cohort study. We considered 132 consecutive eligible adult patients who underwent liver transplantation, attending every three/six months (or more often when needed) from January 1995 to December 2020 and for at least two years after intervention, the “Chronic non-viral hepatitis clinic”, pertaining to the U.O.C. of Internal Medicine, II Polyclinic of Naples, University of Campania “Luigi Vanvitelli”, and the “Liver Transplant Clinic”, pertaining to the Hepatology and Interventional Ultrasound Unit, Gragnano Hospital, ASL Naples 3 South.

The transplantation procedures were conducted at various hospitals spanning across different regions within Italy and, in some cases, even in other European hospital centers, where patients were generally followed for the first year after OLT. The aetiology of cirrhosis was inferred from the liver biopsy performed before OLT and/or the explant liver biopsy report. The causes of liver disease requiring OLT were distributed as follows: 34 (30.9%) viral cirrhosis, 32 (29.1%) non-viral causes (dysmetabolic, alcoholic and/or cryptogenic cirrhosis; hepatocellular carcinoma with and without cirrhosis), and 44 (40%) both viral and non-viral aetiology. All included patients were in the outpatient setting without evidence of acute graft rejection or dysfunction, technical complications, or active infections. Re-transplanted (n = 2) or multi-organ transplanted patients (n = 4), autoimmune hepatitis (n = 1) and hemochromatosis (n = 0) causes of OLT, and subjects with missing data (n = 8) were excluded from the study. In addition, the pre-OLT presenting symptoms suggestive of CVD, the pre-OLT history of documented cardiovascular events or coronary stent implantation (n = 2), as well as the presence of moderate–severe hydrosaline-retention (n = 5) compromising BMI and waist circumference (WC) evaluation were criteria for exclusion from the study. A total of 110 OLT recipient patients were finally included (Figure 1).

All information was obtained from electronic medical records and transferred to a Microsoft Excel table. In addition to age, sex, and cause of transplantation, the following clinical data were collected for each patient: self-reported personal history of pre-OLT angina pectoris, nonfatal MI, nonfatal stroke or symptoms suggestive of CVD; family history of CVD and type 2 diabetes (T2D); current pre/post-OLT smoking and alcohol abuse; pre/post-OLT co-morbidities such as arterial hypertension, T2D and dyslipidaemia with respective medications; anti-rejection therapy; pre/post-OLT measurement of body mass index (BMI), WC and blood values of glucose, triglycerides, cholesterol (total, LDL and HDL), creatinine, and γ-glutamyl transferase (GGT). Pre-OLT and post-OLT data were recorded within one month before surgery and during the last follow-up visit at our two clinics, respectively. For glucose and lipids, blood samples were obtained after an overnight fast. CKD-EPI creatinine formula (2021 update) was used to estimate the glomerular filtration rate (eGFR). Diabetes, arterial hypertension, and dyslipidaemia were diagnosed according to the 2016 European guidelines [28]. The criteria established by the National Cholesterol Education Program were used to diagnose metabolic syndrome (MetS): WC > 102 cm in men and >88 cm in women, Triglycerides levels > 150 mg/dL, HDL < 40 mg/dL in men and <50 mg/dL in woman, blood pressure ≥ 130/≥ 80 or in treatment, fasting glucose ≥ 110 mg/dL [29]. For all patients, the pre- and post-OLT FLI was calculated according to the formula reported by Bedogni et al. (17): FLI = (exp[0.953 × loge (triglycerides) + 0.139 × BMI + 0.718 × loge (GGT + 0.053 × WC − 15.745])/(1 + exp[0.953 × loge (triglycerides) + 0.139 × BMI + 0.718 × loge (GGT) + 0.053 × WC − 15.745]) × 100, where levels of triglycerides are expressed as mmol/L and those of GGT as U/L, BMI calculated with the usual formula (weight in kilograms divided by height in meters squared) and WC measured to the nearest 0.1 cm at the end of a normal expiration in a standing position. The value of pre-OLT FLI was the mean of at least two measurements at the screening examination carried out during the three months preceding the transplantation. The score of FLI ranges from 0 to 100 [20].

The primary outcome of the study was the occurrence in the post-transplantation follow-up of myocardial infarction (MI) (documented instrumentally and/or enzymatically), ischemic stroke, and cardiovascular death. Deaths from non-cardiovascular causes were also registered. Subjects completed the study at the end of the follow-up period or on the date of eventual cardiovascular or non-cardiovascular death. All patients provided written informed consent for data storage and analysis. The study was conducted in accordance with the 1976 Declaration of Helsinki and was approved by the local ethics committees (University of Campania “Luigi Vanvitelli”, Azienda Ospedaliera Universitaria, “Luigi Vanvitelli”, Azienda Ospedaliera di Rilievo Nazionale “Ospedale dei Colli”; approval ID. 12771/I; approval date 2 May 2023). Clinical trial.gov registration ID NCT05895669.

### Statistical Analysis

The entire dataset was initially analyzed using descriptive statistical indices. For a comprehensive understanding, qualitative data were represented in terms of absolute and relative percentage frequencies. Conversely, for quantitative continuous data, we employed two different descriptive methods based on data distribution characteristics. Specifically, normally distributed data were described using the mean and standard deviation (SD), whereas non-normally distributed data were summarized using the median and interquartile range (IQR). To determine the normality of the data distribution, the Shapiro–Wilk test was conducted prior to further analysis.

To assess the predictive value of the pre-OLT FLI for the occurrence of post-OLT cardiovascular events, we conducted a Receiver Operating Characteristic (ROC) curve analysis. This analysis aimed to identify a potential cutoff value for the FLI that could effectively discriminate between patients at risk and those not at risk of cardiovascular events following transplantation. Based on the outcome of this analysis, participants were categorized into two distinct groups: those with FLI values above the determined cutoff and those with FLI values equal to or below the cutoff. Subsequently, we explored the differences between these two subgroups through appropriate statistical tests. For the analysis of qualitative variables, we employed either the Chi-squared test or, when applicable, the Fisher–Freeman–Haltob exact test. Continuous data, on the other hand, were compared using either the Student’s t-test or the Mann–Whitney U test, depending on the underlying data distribution. To gauge the accuracy of our predictive model, we employed Harrel’s C index, which is equivalent to the ROC curve and ranges from 0 to 1. Higher values of this index signify greater accuracy in predicting outcomes.

Our primary endpoint, defined as the first-event free survival following liver transplantation, was assessed using a Kaplan–Meier survival analysis. We employed the log-rank test to compare survival curves and determine if there were statistically significant differences among different groups.

Furthermore, Kaplan–Meier curves were generated to visually depict the survival estimates over time. To identify potential prognostic factors, whether negative or positive, that could influence survival outcomes, we utilized univariable and multivariable Cox proportional hazards regression models. These models provided us with hazard ratios (HR) and their corresponding 95% confidence intervals (CI), enabling us to assess the strength and significance of these factors in relation to the primary endpoint.

Statistical analyses were conducted using SPSS statistical software (version 24.0, SPSS, Chicago, IL, USA) and STATA 14.0 software (StataCorp, College Station, TX, USA).

## 3. Results

We identified 110 eligible Caucasian patients (median age 57 years [IQR: 50–62], 72.7% male) who were followed for a median of 92.3 months (IQR: 45.7–172.4) after liver transplantation.

The patient characteristics at pre-OLT and at the last post-OLT ambulatory visit are detailed in Table 1. The comparison between the two times showed a significantly higher prevalence of metabolic syndrome (MetS), type 2 diabetes (T2D), arterial hypertension, dyslipidaemia, as well as significantly higher blood levels of triglycerides, total cholesterol, LDL cholesterol, and HDL cholesterol. However, lower levels of GGT and estimated glomerular filtration rate (eGFR) were observed at the last follow-up visit. No significant differences were found for body mass index (BMI), waist circumference (WC), impaired fasting glucose, or FLI value. The prevalence of tobacco habit and alcohol abuse was higher at the pre-OLT stage. Harrel’s C index for assessing the association between the FLI and cardiovascular outcomes was found to be 0.700 (CI 95% 0.599 to 0.801).

The receiver operating characteristic (ROC) analysis estimated a cutoff value of 66.0725 for predicting post-OLT cardiovascular events, with an 86.7% sensitivity and a 63.7% specificity (Figure 2). The area under the curve (AUC) for FLI in predicting post-OLT cardiovascular events was 0.7. Based on this cutoff, participants were classified into two groups: FLI ≤ 66 or FLI > 66.

At the pre-OLT stage, 48 out of 110 patients had FLI > 66 (Table 2), with higher BMI, WC, prevalence of obesity and overweight, and higher GGT levels when compared to the 62 patients with FLI ≤ 66. No differences were observed for the other parameters. As depicted in Table 3, there was no difference in median follow-up duration between patients with FLI ≤ 66 and those with FLI > 66. At the last clinic visit, patients with FLI > 66 exhibited higher BMI, WC, prevalence of obesity and overweight, as well as a significantly higher prevalence of MetS and a higher median FLI value compared to patients with FLI ≤ 66.

During the follow-up period, 16 patients (14.5%) developed at least one adverse cardiovascular event. Among them, three experienced both non-fatal myocardial infarction (MI) and ischemic stroke. Despite the small number of events, the incidence of fatal and non-fatal stroke, particularly fatal and non-fatal MI, was significantly higher in the FLI > 66 group (27.1% vs. 4.8%; *p* = 0.001) (Table 4). Five patients died, with two deaths attributed to cardiovascular disease and three to non-cardiovascular causes (massive pulmonary bleeding due to Rendu-Osler Syndrome, complications during intensive care unit stay for COVID-19 infection, hepatocellular carcinoma relapse).

The Kaplan–Meier analysis revealed a worse cardiovascular event-free survival in patients with FLI > 66 compared to those with FLI ≤ 66 (log-rank: 0.0008) (Figure 3).

The univariate Cox regression analysis performed for all pre-OLT parameters showed that age, FLI > 66, BMI, WC, obesity, smoking, and history of CVD before transplantation were significantly associated with the incidence of new-onset cardiovascular events (Table 5).

Considering the limited number of recorded cardiovascular events, only variables that showed significance in the univariate analysis were included in the multivariate analysis. Additionally, BMI and WC were not considered to be these parameters already included in the calculation of FLI and obesity was already represented by these measures. Ultimately, the two variables independently associated with cardiovascular risk were FLI > 66 and smoking habits at the pre-OLT stage, while the role of age and pre-OLT CVD was not significant (Table 6).

## 4. Discussion

The present cohort study investigated, for the first time, the relationship between the FLI, a well-validated surrogate marker of nonalcoholic fatty liver disease (NAFLD), and the risk of cardiovascular events in a population of individuals who underwent liver transplantation. The main finding of this investigation was the significant independent predictive value of FLI for the long-term post-OLT occurrence of the composite outcome of MI, ischemic stroke, and cardiovascular mortality, irrespective of age and personal history of previous cardiovascular events. Specifically, the study identified an FLI cutoff value > 66 as an accurate indicator of higher cardiovascular risk in the late post-transplantation period, with an incidence of cardiovascular events 5.5 times higher than in patients with FLI ≤ 66. This FLI cutoff value may serve as an appropriate threshold for non-invasive stress testing before liver transplantation in patients without cardiovascular symptoms and for enhanced surveillance of cardio-metabolic risk following OLTm facilitating early intervention for preventive purposes.

Our FLI cutoff value closely corresponds to the previously established threshold of ≥60, documented in this study as a reliable predictor of arterial hypertension in individuals who have not yet been exposed to antihypertensive medications [30]. The same FLI threshold has also been associated with significant clinical conditions, such as the onset of left ventricular remodelling and diastolic dysfunction [31,32]. Of noteworthy significance is a recent extensive population-based cohort investigation that may underscore the clinical relevance of our findings [33]. In this study, persistent NAFLD, as defined by a FLI score of ≥60, emerged as a potent predictor of adverse outcomes such as all-cause mortality, MI, and stroke when compared to those with either no NAFLD or sporadic, intermittent NAFLD [33].

Patients who are diagnosed with end-stage liver disease are often thought to possess a relatively improved cardiovascular risk profile. This presumption stems from several key peculiarities of this type of subject. First, liver insufficiency reduces the production of cholesterol, which notoriously plays a crucial role in the development of atherosclerosis. Second, these patients frequently exhibit peripheral vasodilation coupled with normal to lower arterial blood pressure, contributing to a decreased risk of cardiovascular events. Additionally, increased estrogen levels, commonly observed in this population, are believed to exert protective effects against atherosclerotic plaque formation, further suggesting a reduced susceptibility to cardiovascular complications [34,35,36]. However, it is worth noting that despite these apparent protective mechanisms, the prevalence of coronary heart disease in patients awaiting liver transplantation is unexpectedly higher than anticipated, as evidenced by various studies [37,38]. Furthermore, once liver transplantation has taken place, coronary heart disease emerges as a substantial contributor to post-transplant morbidity and mortality [39]. These findings align well with the observations made in our study, underscoring the importance of the cardiovascular risk in patients with end-stage liver disease that makes the cardiovascular assessment an essential component of pre-OLT evaluation despite controversies about the methodological approach to be used.

Given the substantial impact of OLT on cardiovascular mortality, especially over the long term [5], the availability of a novel risk stratification tool, derived from a mere quartet of straightforward clinical variables readily obtainable in any healthcare or nursing setting, holds significant promise in mitigating the medical and economic burden associated with the screening and diagnosis of CVD within this patient population. The intriguing revelation from our study is the predictive efficacy of the FLI value assessed prior to transplantation, independently of advancing age and escalating burden of cardiovascular risk factors developed in the post-OLT. This highlights the potential utility of the FLI as a valuable tool for early risk assessment in liver transplant candidates, offering a means to proactively identify individuals at heightened risk of cardiovascular complications in advance of the transplantation procedure and potentially enabling more targeted and cost-effective interventions to safeguard the cardiovascular health of these patients.

It is well known that FLI encompasses some traditional cardiovascular risk factors such as BMI, triglyceride levels, and WC. Thus, the resulting index value may reflect the worsening of these variables, specifically BMI. Nevertheless, the intricate interplay between the pathophysiological mechanisms underpinning hepatic steatosis and atherosclerosis must not be neglected. NAFLD demonstrates a close association with the hallmarks of metabolic syndrome, encompassing insulin resistance, lipotoxicity, oxidative stress, and altered adipocytokine secretion. These factors collectively contribute to a condition of chronic low-grade inflammation, which in turn leads to the simultaneous accumulation of fat in the liver and the formation of atheroma in the intimal layer of the arterial wall. Intriguingly, NAFLD has been demonstrated to elevate the risk of vascular endothelial dysfunction and the progression of atherosclerosis, even independently of metabolic syndrome and its individual components [40,41]. In this context, numerous pieces of evidence emphasize the main role of endothelial dysfunction as a common pathogenic factor in NAFLD and NAFLD-associated atherosclerosis [42].

In addition to pre-OLT FLI, the pre-OLT smoking habit also has been shown to independently predict the long-term cardiovascular risk in adult OLT recipients we studied, being associated with a threefold increase in the risk of cardiovascular events, despite the fact that the majority of patients stopped tobacco use after surgery. Notably, the prevalence of tobacco use among liver transplant recipients is noteworthy, with reported rates spanning from 14.7% to as high as 75% [43]. However, it must be admitted that our comprehension of the potential implications of cigarette smoking on morbidity and mortality following liver transplantation remains relatively underexplored within the realm of research. As such, a significant knowledge gap persists regarding the intricate relationship between cigarette smoking and health outcomes post-liver transplantation, warranting comprehensive investigation and scrutiny. This finding, though intriguing, poses a challenge in terms of interpretation, particularly in light of the limited literature available for comparison. The existing body of research presents varying perspectives on the association between smoking and post-OLT outcomes. For instance, one study found no independent link between active smoking one year after liver transplantation and vascular events [44]. By contrast, in a separate study, individuals who were active smokers at the time of listing for OLT procedures were found to have an independent association with the development of cardiovascular disease in the post-OLT period, with a hazard ratio of 10.91 (95% confidence interval 1.22–97.92) [45]. Conversely, another study has presented evidence indicating that active smokers presented a 79% higher risk of mortality than those who had never smoked or quit smoking before OLT [46].

Cardiovascular disease is reported as a complication of chronic immunosuppression, which can negatively affect long-term survival and quality of life in liver transplantation [8]. Due to the very different extension of follow-up in our patients, multiple combination regimens and newly introduced drugs may have occurred over the years, making it difficult to adequately evaluate the eventual impact of various immunosuppressive agents on CVD [47,48].

In conclusion, this study emphasizes the critical importance of assessing cardiovascular risk in liver transplant candidates, given the absence of a standardized method. The FLI emerges as a valuable screening tool, predicting potentially fatal cardiovascular events. Its consistency suggests potential clinical utility, although integration into routine assessments must be approached cautiously. Further validation through extensive, multicenter studies is crucial to establish its reliability across diverse populations.

Future research should explore FLI-guided interventions for hepatic steatosis and investigate smoking’s specific impact on cardiovascular health in liver transplant recipients. While FLI shows promise, its clinical application requires a careful, evidence-driven approach.

The quest for improved cardiovascular care and outcomes in liver transplant candidates continues to be a complex and evolving challenge, demanding further investigation and innovation in the field.

### Limitations

The study suffers from various limitations that need to be acknowledged. Firstly, its retrospective design inherently carries important biases. Secondly, the sample size was relatively modest, which may have influenced the statistical power to detect certain associations, as well as the number of recorded cardiovascular events during the follow-up period was relatively low. The Caucasian race of our whole population limits the generalizability of the findings to other ethnic groups. Furthermore, our reliance on self-reported information regarding the history or status of pre-transplant cardiovascular disease introduces potential recall bias and may not capture all relevant clinical details accurately. Additionally, exhaustive information on smoking habits was not available, which could affect the precision of our analysis, as smoking is a well-known complex and multifactorial risk factor for cardiovascular events. Finally, we could not manage to assess the impact of immunosuppressive treatment on cardiovascular outcomes due to the variations in the duration of follow-up periods and the utilization of diverse combination regimens and modifications.

## Figures and Tables

**Figure 1 biomedicines-11-02866-f001:**
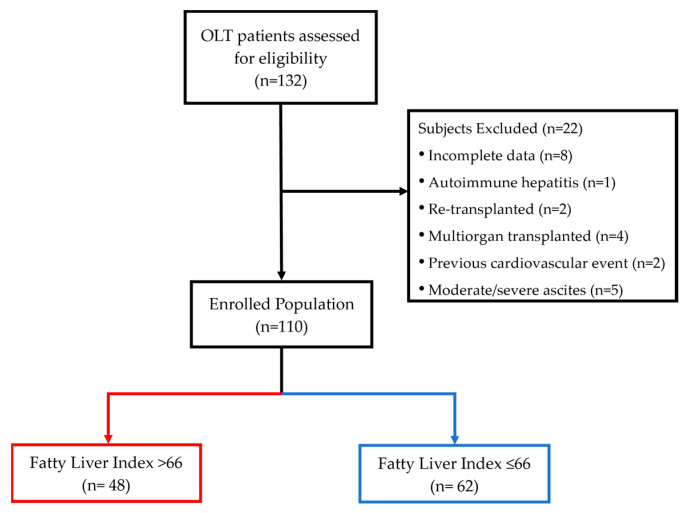
Study flow-chart.

**Figure 2 biomedicines-11-02866-f002:**
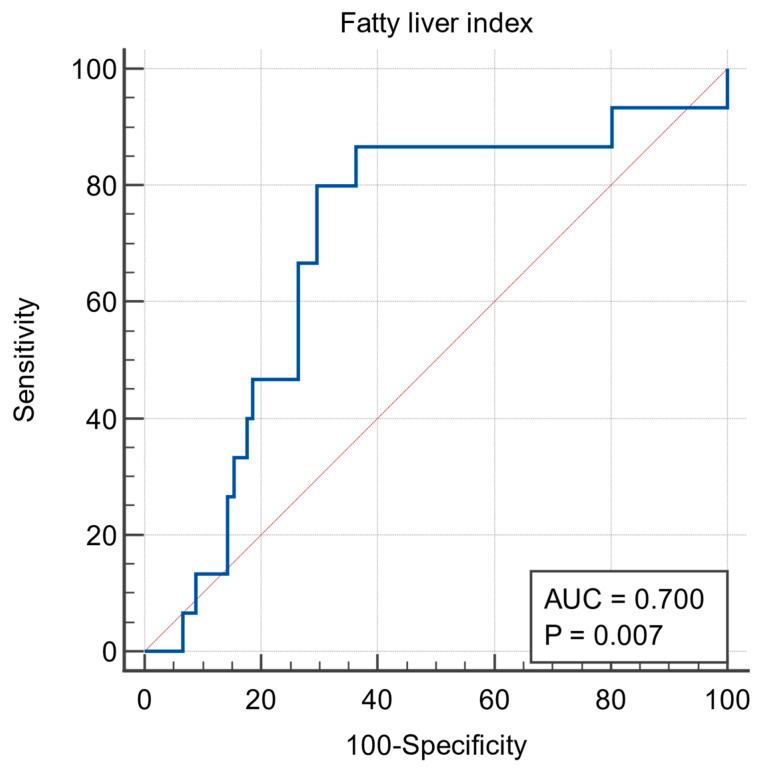
ROC curve for the definition of the fatty liver index cut-off value (sensitivity: 86.7%, specificity 63.7%).

**Figure 3 biomedicines-11-02866-f003:**
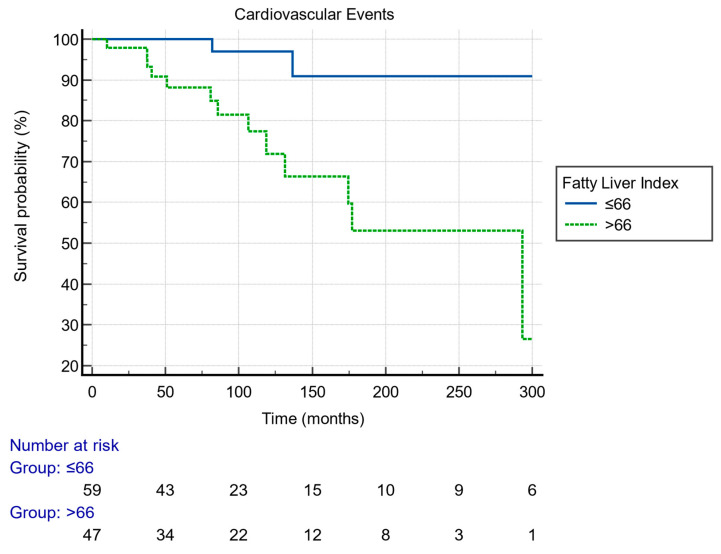
Kaplan–Meier survival analysis estimating the risk of cardiovascular events among OLT patients according to FLI.

**Table 1 biomedicines-11-02866-t001:** Pre- and post-OLT characteristics of the study sample.

Parameter	Baseline (n = 110)	Follow-Up (n = 110)	*p*
**Age**, years, median [IQR]	57.0 [50.0–62.0]	67.0 [61.0–72.0]	**<0.001**
**Sex**, n (%)			
*M*	80 (72.7)		
*F*	30 (27.3)		
**FLI**, median [IQR]	59.5 [38.3–82.3]	55.3 [33.4–79.1]	0.324
**BMI**, kg/m^2^, median [IQR]	26.6 [23.5–28.4]	26.5 [23.9–30.0]	0.505
**Obese**, n (%)	17 (16.0)	18 (16.4)	*0.098*
**Overweight**, n (%)	50 (45.5)	55 (55.0)	0.137
**Waist circumference**, cm, median [IQR]	103.0 [90.0–113.6]	100.0 [92.0–115.0]	0.893
**Impaired fast glucose**, n (%)	7 (6.6)	5 (4.5)	0.553
**Diabetes**, n (%)	15 (14.2)	43 (39.1)	**0.001**
**Hypertension**, n (%)	16 (14.5)	79 (71.8)	**<0.001**
**Total cholesterol**, mg/dL, median [IQR]	123.5 [105.0–165.5]	192.5 [160.0–225.0]	**<0.001**
**LDL**, mg/dL, median [IQR]	86.5 [66.0–105.0]	128.0 [100.0–155.0]	**<0.001**
**HDL**, mg/dL, median [IQR]	33.5 [28.0–44.0]	48.0 [39.0–59.5]	**<0.001**
**Triglycerides**, mg/dL, median [IQR]	89.0 [71.0–118.0]	123.0 [86.0–167.0]	**<0.001**
**Dyslipidemia**, n (%)	10 (9.1)	68 (63)	**<0.001**
**Metabolic Syndrome**, n (%)	15 (13.6)	44 (40)	**<0.001**
**GGT**, U/L, median [IQR]	57.5 [42.0–84.0]	28.5 [17.5–53.0]	**<0.001**
**Smoking**, n (%)	34 (32.1)	13 (11.8)	**0.0002**
**Alcohol abuse**, n (%)	24 (21.8)	2 (1.8)	**<0.001**
**eGFR**, mL/min/m^2^, median [IQR]	91.9 [76.0–103.8]	69.2 [55.3–90.4]	**<0.001**

OLT: orthotopic liver transplantation; IQR: interquartile range; FLI: fatty liver index; BMI: body mass index; GGT: γ-glutamyltransferase; eGFR: estimated glomerular filtration rate.

**Table 2 biomedicines-11-02866-t002:** Pre-OLT characteristics of the study sample categorized using the fatty liver index.

Parameter	Overall (n = 110)	FLI ≤ 66 (n = 62)	FLI > 66 (n = 48)	*p*
**Age**, years, median [IQR]	57.0 [50.0–62.0]	56.5 [48.0–63.0]	57.0 [52.0–60.0]	0.845
**Sex**, n (%)				*0.079*
*M*	80 (72.7)	41 (66.1)	39 (81.3)	
*F*	30 (27.3)	21 (33.9)	9 (18.8)	
**Family history of CVD**, n (%)	10 (9.3)	3 (4.8)	7 (14.6)	0.095
**Family history of diabetes**, n (%)	12 (11.2)	3 (4.8)	9 (18.8)	0.144
**Personal history of MI**, n (%)	2 (1.8)	0	2 (4.2)	0.106
**Personal history of Stroke**, n (%)	3 (2.7)	2 (3.2)	1 (2.1)	0.716
**FLI**, median [IQR]	59.5 [38.3–82.3]	41 [26.3–56.6]	89.0 [74.6–95.3]	**<0.001**
**BMI**, kg/m^2^, median [IQR]	26.6 [23.5–28.4]	24.5 [22.7–26.0]	27.8 [27.0–31.8]	**<0.001**
**Obese**, n (%)	17 (16.0)	0	17 (35.4)	**<0.001**
**Overweight**, n (%)	50 (45.5)	21 (33.9)	29 (60.4)	**0.008**
**Waist circumference**, cm, median [IQR]	103.0 [90.0–113.6]	92.0 [87.0–100.0]	116 [107.8–128.0]	**<0.001**
**Impaired fast glucose**, n (%)	7 (6.6)	4 (6.5)	3 (6.3)	0.935
**Diabetes**, n (%)	15 (14.2)	9 (14.5)	6 (12.5)	0.716
**Hypertension**, n (%)	16 (14.5)	10 (16.1)	6 (12.5)	0.594
**Total cholesterol**, mg/dL, median [IQR]	123.5 [105.0–165.5]	119.0 [103.0–152.0]	142.0 [116.8–185.8]	**0.032**
**LDL**, mg/dL, median [IQR]	86.5 [66.0–105.0]	76.0 [58.5–100.3]	99.0 [72.3–119.8]	*0.086*
**HDL**, mg/dL, median [IQR]	33.5 [28.0–44.0]	33.0 [28.0–36.8]	36.0 [25.8–47.0]	0.482
**Triglycerides**, mg/dL, median [IQR]	89.0 [71.0–118.0]	88.5 [69.0–102.0]	88.5 [70.0–142.0]	0.146
**Dyslipidemia**, n (%)	10 (9.1)	5 (8.1)	5 (10.4)	0.672
**Metabolic Syndrome**, n (%)	15 (14.0)	5 (8.1)	10 (20.8)	*0.068*
**GGT**, U/L, median [IQR]	57.5 [42.0–84.0]	51.0 [32.0–70.0]	63.5 [55.0–89.0]	**0.004**
**Smoking**, n (%)	34 (32.1)	21 (33.9)	13 (27.1)	0.387
**Alcohol abuse**, n (%)	24 (21.8)	16 (25.8)	8 (16.7)	0.252
**eGFR**, mL/min/m^2^, median [IQR]	91.9 [76.0–103.8]	91.0 [81.0–104.0]	93.7 [73.3–102.5]	0.668

OLT: orthotopic liver transplantation; IQR: interquartile range; CVD: cardiovascular disease; MI: myocardial infraction; FLI: fatty liver index; BMI: body mass index; GGT: γ-glutamyltransferase; eGFR: estimated glomerular filtration rate.

**Table 3 biomedicines-11-02866-t003:** Post-OLT characteristics of the study sample categorized using the fatty liver index.

Parameter	Overall (n = 110)	FLI ≤ 66 (n = 62)	FLI > 66 (n = 48)	*p*
**Age**, years, median [IQR]	67.0 [61.0–72.0]	67.0 [60.8–71.3]	67.5 [62.5–73.5]	0.490
**Follow-up months**, median [IQR]	92.3 [45.6–172.4]	89.4 [45.7–172.4]	112.5 [54.7–214.8]	0.357
**FLI**, median [IQR]	55.3 [33.4–79.1]	41.8 [26.5–63.1]	75.0 [48.7–87.6]	**<0.001**
**BMI**, kg/m^2^, median [IQR]	26.5 [23.9–30.0]	24.9 [23.4–27.4]	28.5 [26.0–32.9]	**<0.001**
**Obese**, n (%)	18 (16.4)	0	18 (37.5)	**<0.001**
**Overweight**, n (%)	55 (55.0)	25 (40.3)	30 (62.5)	**0.022**
**Waist circumference**, cm, median [IQR]	100.0 [92.0–115.0]	95.0 [88.0–100.0]	113.0 [103.5–125.5]	**<0.001**
**Impaired fast glucose**, n (%)	5 (4.5)	3 (4.8)	2 (4.2)	0.867
**Diabetes**, n (%)	43 (39.1)	22 (35.5)	21 (43.8)	0.380
**Hypertension**, n (%)	79 (71.8)	46 (74.2)	33 (68.8)	0.531
**Total cholesterol**, mg/dL, median [IQR]	192.5 [160.0–225.0]	190.0 [160.0–225.0]	194.0 [153.0–225.0]	0.939
**LDL**, mg/dL, median [IQR]	128.0 [100.0–155.0]	128.0 [97.0–151.0]	128.0 [106.0–156.0]	0.662
**HDL**, mg/dL, median [IQR]	48.0 [39.0–59.5]	49.0 [41.0–62.5]	46.0 [34.5–58.5]	0.144
**Triglycerides**, mg/dL, median [IQR]	123.0 [86.0–167.0]	112.0 [81.0–158.0]	129.5 [96.0–182.0]	0.128
**Dyslipidemia**, n (%)	68 (63)	36 (58.1)	32 (66.7)	0.336
**Metabolic Syndrome**, n (%)	44 (41.1)	19 (30.6)	25 (52.1)	**0.039**
**GGT**, U/L, median [IQR]	28.5 [17.5–53.0]	29.0 [16.0–53.0]	28.0 [18.0–46.0]	0.763
**Smoking**, n (%)	13 (11.8)	8 (12.9)	5 (10.4)	0.690
**Alcohol abuse**, n (%)	2 (1.8)	0	2 (4.2)	0.106
**eGFR**, mL/min/m^2^, median [IQR]	69.2 [55.3–90.4]	73.1 [60.2–91.9]	65.5 [51.5–80.4]	*0.078*

OLT: orthotopic liver transplantation; IQR: interquartile range; FLI: fatty liver index; BMI: body mass index; GGT: γ-glutamyltransferase; eGFR: estimated glomerular filtration rate.

**Table 4 biomedicines-11-02866-t004:** Post-OLT cardiovascular events and overall mortality in the study sample categorized using FLI.

Parameter	Overall (n = 110)	FLI ≤ 66 (n = 62)	FLI > 66 (n = 48)	*p*
**Incident fatal and non-fatal MI**, n (%)	16 (14.5)	3 (4.8)	13 (27.1)	**0.001**
**Incident fatal and non-fatal Stroke**, n (%)	3 (2.8)	0	3 (6.3)	**0.047**
**Cardiovascular death**, n (%)	2 (2.2)	1 (1.6)	1 (2.1)	0.739
**Overall incident CV events**, n (%)	16 (14.5)	3 (4.8)	13 (27.1)	**0.001**
**Overall mortality**, n (%)	5 (5.4)	2 (3.2)	3 (6.3)	**0.663**

MI: myocardial infarction; CV: cardiovascular.

**Table 5 biomedicines-11-02866-t005:** Univariable Cox regression model.

	Univariable Analysis			
Parameter	HR	95% CI		*p*
**Age**	1.07	1.00	1.15	**0.049**
**Sex**				
M (ref)				
F		0.09	1.25	0.104
**Family history of MI**	1.59	0.70	3.61	0.267
**Family history of diabetes**	1.52	0.34	6.78	0.580
**Personal history of MI**	12.29	1.36	110.80	**0.025**
**Personal history of Stroke**	0.00	0.00	inf	0.960
**Fatty liver index**				
**≤66**	1			
**>66**	6.34	1.78	22.56	**0.004**
**BMI**	1.13	1.03	1.24	**0.011**
**Waist circumference**	1.05	1.02	1.08	**0.002**
**Obese**	5.02	1.38	18.32	**0.015**
**Overweight**	1.25	0.47	3.39	0.648
**Impaired fast glucose**	1.19	0.15	9.20	0.866
**Diabetes**	1.83	0.40	8.40	0.439
**Hypertension**	0.70	0.09	5.35	0.728
**Total cholesterol**	0.99	0.98	1.02	0.996
**LDL**	0.99	0.96	1.03	0.811
**HDL**	1.05	0.95	1.16	0.313
**Triglycerides**	0.99	0.99	1.01	0.976
**Dyslipidemia**	2.84	0.81	9.97	0.104
**Metabolic Syndrome**	0.48	0.06	3.65	0.480
**GGT**	0.99	0.98	1.01	0.729
**Smoking**	3.00	1.06	8.49	**0.038**
**Alcohol Abuse**	1.72	0.55	5.35	0.350
**eGFR**	1.01	0.98	1.04	0.482

MI: myocardial infraction; BMI: body mass index; GGT: γ-glutamyltransferase; eGFR: estimated glomerular filtration rate.

**Table 6 biomedicines-11-02866-t006:** Multivariable Cox regression model.

	Multivariable Analysis			
Parameter	HR	95% CI		*p*
**Age**	1.06	0.97	1.15	0.185
**Fatty liver index**				
**≤66**	1			
**>66**	5.50	0.51	59.85	**0.010**
**Smoking**	3.20	1.01	10.12	**0.048**
**Personal history of MI**	3.04	0.31	30.17	0.343

MI: myocardial infraction.

## Data Availability

The data that support the findings of this study are available upon reasonable request from the corresponding author.

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
