# Peer review of "Predictive Value of Fatty Liver Index for Long-Term Cardiovascular Events in Patients Receiving Liver Transplantation: The COLT Study"

_biomedicines, 2023, doi:10.3390/biomedicines11102866_

Round 1
Reviewer 1 Report
Fatty Liver Index (FLI) has been reported to be strongly associated with carotid and coronary atherosclerosis, as well as cardiovascular mortality. This study further reported that the significant independent predictive value of FLI for long-term post- orthotopic liver transplantation (OLT) occurrence of myocardial infarction (MI) and ischemic stroke, irrespective of age and personal history of previous cardiovascular events. Specifically, this study identified a FLI cutoff value >66 as an accurate indicator of higher cardiovascular risk in the late post-transplantation period, with an incidence of cardiovascular events 5.5 times higher than in patients with FLI ≤66. There are some concerns as listed in the following:
(1) As a discussion in the Discussion section (L370-375) concerning about that cardiovascular disease is reported as a complication of chronic immunosuppression, it is better to give some information about the Post-OLT medication managements for these patients in the present study.
(2) Typos and others:
*L38-39: The keywords myocardial infraction and stroke did not appear in the Abstract. It is better to include them into the Abstract.
*L95: FLI full name first
*L225: Figure 1 AUC=0,700. P=0,007 -> AUC=0.700. P=0.007
*L228: (Table II) -> (Table 2)
*L231: Table III -> Table 3
*L240: Table IV -> Table 4
*L254: Table 4: MI myocardial infraction?
L266: (Table V) -> (Table 5)
L277: (Table VI) -> (Table 6)
**L284: The main finding of investigation was the significant independent predictive value of FLI for long-term post-OLT occurrence of myocardial infarction (MI), 285 ischemic stroke, and cardiovascular mortality -> For cardiovascular mortality, it shows no significance (P-0.793 in Table 4)
*L295: documented in the by a study?
*L430: References: check all to keep consistent writing format for the title (capital prefix only on the first word vs. on all words) and page number (243-253 vs. 243-53).
L442: 6(4):243-253 -> 6(4):243-53
L445: Re-Evaluating Age Limits in Transplantation (REALT)… -> Re-evaluating age limits in transplantation (REALT)…
L447: 1492-1503 -> 1492-503
L448: Trends in Liver Disease Etiology…
L463: Incidence of and Risk Assessment…
L464: 1645-1657 -> 1645-57
L466: The Role of Arterial Stiffness…
L470: 1968-1979 -> 1968-79
L474: Dysregulated Epicardial Adipose Tissue…
L478: 833-839 -> 833-9
L481: Effect on Survival…
L487: 755-768 -> 755-68
L489: Non-Alcoholic Fatty Liver Disease…
L496: Relationship Among Fatty Liver..
L497: 1453-1463-> 1453-63
L507: 2315-2381 -> 2315-81
L509: Expert Panel on Detection…
L513: 119-127 -> 119-27
L516: 961-969 -> 961-9
L518: Independent Association of Fatty Liver Index…
L519: 139-146 -> 139-46
L520: The Repeatedly Elevated Fatty Liver Index..
L536: 875-885 -> 875-85
L542: 6651-6666 -> 6651-66
L543: Endothelial Cell Dysfunction…
L552: Active Smoking Before Liver Transplantation…
L557: 5155-5163-> 5155-63
Author Response
Fatty Liver Index (FLI) has been reported to be strongly associated with carotid and coronary atherosclerosis, as well as cardiovascular mortality. This study further reported that the significant independent predictive value of FLI for long-term post- orthotopic liver transplantation (OLT) occurrence of myocardial infarction (MI) and ischemic stroke, irrespective of age and personal history of previous cardiovascular events. Specifically, this study identified a FLI cutoff value >66 as an accurate indicator of higher cardiovascular risk in the late post-transplantation period, with an incidence of cardiovascular events 5.5 times higher than in patients with FLI ≤66. There are some concerns as listed in the following:
- As a discussion in the Discussion section (L370-375) concerning about that cardiovascular disease is reported as a complication of chronic immunosuppression, it is better to give some information about the Post-OLT medication managements for these patients in the present study.
R. Thank you for your insightful feedback and valuable suggestions. We sincerely appreciate your thorough review of our work. We regret to inform you that, as mentioned in the subsequent sentence in the text, it was not feasible to assess the impact of immunosuppressive treatment, despite its significant relevance. We acknowledge the importance of this aspect and fully recognize the interest it holds for both the scientific community and the readers. Rest assured; we are committed to addressing this specific topic in our future studies. We understand the importance of exploring the effects of immunosuppressive treatment on cardiovascular outcomes and will make it a priority in our upcoming research endeavors. We have added to the limitation of the study the following sentence “Finally, we could not manage to assess the impact of immunosuppressive treatment on cardiovascular outcomes, due to the variations in the duration of follow-up periods and the utilization of diverse combination regimens and modifications.”
(2) Typos and others:
*L38-39: The keywords myocardial infraction and stroke did not appear in the Abstract. It is better to include them into the Abstract.
R. We have modified accordingly.
*L95: FLI full name first
R. We have modified accordingly.
*L225: Figure 1 AUC=0,700. P=0,007 -> AUC=0.700. P=0.007
R. We have modified accordingly.
*L228: (Table II) -> (Table 2)
R. We have modified accordingly.
*L231: Table III -> Table 3
R. We have modified accordingly.
*L240: Table IV -> Table 4
R. We have modified accordingly.
*L254: Table 4: MI myocardial infraction?
R. We wish to apologize with the reviewer, but some of the tables were corrupted during the process. MI was the table legend. We have modified the table.
L266: (Table V) -> (Table 5)
R. We have modified accordingly.
L277: (Table VI) -> (Table 6)
R. We have modified accordingly.
**L284: The main finding of investigation was the significant independent predictive value of FLI for long-term post-OLT occurrence of myocardial infarction (MI), 285 ischemic stroke, and cardiovascular mortality -> For cardiovascular mortality, it shows no significance (P-0.793 in Table 4)
R. We thank the reviewer for his/her precious comments. We have modified the text as follows: “The main finding of investigation was the significant independent predictive value of FLI for long-term post-OLT occurrence of the composite outcome of MI, ischemic stroke, and cardiovascular mortality, irrespective of age and personal history of previous cardiovascular events.”
*L295: documented in the by a study?
R. We modified as follows: documented in this study.
*L430: References: check all to keep consistent writing format for the title (capital prefix only on the first word vs. on all words) and page number (243-253 vs. 243-53).
L442: 6(4):243-253 -> 6(4):243-53
R. Done
L445: Re-Evaluating Age Limits in Transplantation (REALT)… -> Re-evaluating age limits in transplantation (REALT)…
R. Done
L447: 1492-1503 -> 1492-503
R. Done
L448: Trends in Liver Disease Etiology…
R. Done
L463: Incidence of and Risk Assessment…
R. Done
L464: 1645-1657 -> 1645-57
R. Done
L466: The Role of Arterial Stiffness…
R. Done
L470: 1968-1979 -> 1968-79
R. Done
L474: Dysregulated Epicardial Adipose Tissue…
R. Done
L478: 833-839 -> 833-9
R. Done
L481: Effect on Survival…
R. Done
L487: 755-768 -> 755-68
R. Done
L489: Non-Alcoholic Fatty Liver Disease…
R. Done
L496: Relationship Among Fatty Liver..
R. Done
L497: 1453-1463-> 1453-63
R. Done
L507: 2315-2381 -> 2315-81
R. Done
L509: Expert Panel on Detection…
R. Done
L513: 119-127 -> 119-27
R. Done
L516: 961-969 -> 961-9
R. Done
L518: Independent Association of Fatty Liver Index…
R. Done
L519: 139-146 -> 139-46
R. Done
L520: The Repeatedly Elevated Fatty Liver Index..
R. Done
L536: 875-885 -> 875-85
R. Done
L542: 6651-6666 -> 6651-66
R. Done
L543: Endothelial Cell Dysfunction…
R. Done
L552: Active Smoking Before Liver Transplantation…
R. Done
L557: 5155-5163-> 5155-63
R. Done
Reviewer 2 Report
For author
Reviewing the manuscript entitled, “Predictive value of fatty liver index for long-term cardiovascular events in patients receiving liver transplantation: the COLT study” by Caturano A et al., this is a retrospective cohort study about relationship between FLI and cardiovascular event in patients who were performed OLT. Since FLI is a marker that can be obtained through simple calculations, I believe that the obtained results will have practical clinical benefits. So, the authors need to respond to my following concerns.
Although there is a description at the beginning of the results regarding the underlying diseases of OLT patients, but this should be moved to 2. Patients and Methods. And the authors should attach a flowchart diagram regarding patient eligibility.
I think that OLT is the only therapeutic option for patients with end-stage liver disease. For example, if a patient is in the end stage of liver cancer, it is unlikely that he would be overnutrition state such as DM, overweight. If so, would it be important to sub-analyze the underlying disease of OLT surgery?
The authors need to modify table 1. This is blurred.
Limitations are fine. For OLT patients, the number of cases is limited.
There is almost no problem, but some editing is required.
Author Response
Reviewing the manuscript entitled, “Predictive value of fatty liver index for long-term cardiovascular events in patients receiving liver transplantation: the COLT study” by Caturano A et al., this is a retrospective cohort study about relationship between FLI and cardiovascular event in patients who were performed OLT. Since FLI is a marker that can be obtained through simple calculations, I believe that the obtained results will have practical clinical benefits. So, the authors need to respond to my following concerns.
R. Thank you for your insightful feedback and valuable suggestions. We sincerely appreciate your thorough review of our work.
Although there is a description at the beginning of the results regarding the underlying diseases of OLT patients, but this should be moved to 2. Patients and Methods. And the authors should attach a flowchart diagram regarding patient eligibility.
R. We have modified accordingly. We have also made the study flow-chart diagram as suggested.
I think that OLT is the only therapeutic option for patients with end-stage liver disease. For example, if a patient is in the end stage of liver cancer, it is unlikely that he would be overnutrition state such as DM, overweight. If so, would it be important to sub-analyze the underlying disease of OLT surgery?
R. We sincerely appreciate the thoughtful feedback provided by the reviewer. We wholeheartedly agree that the diverse nature of diseases leading to transplantation results in unique post-OLT comorbidity trajectories. In our study, out of the 32 patients undergoing liver transplantation due to non-viral causes, only 7 cases were attributed to HCC. Regrettably, owing to the limited sample size of this specific patient subset, as well as the overall study cohort, conducting a statistically robust sub-analysis proves unfeasible. Nevertheless, we genuinely value the reviewer's input and will take it into careful consideration for any future endeavors. Thank you for your understanding and support.
The authors need to modify table 1. This is blurred.
R. We have modified the table accordingly.
Limitations are fine. For OLT patients, the number of cases is limited.
R. We thank the reviewer for his/her precious comments. We have reported this issue in the limitation of the study.
Reviewer 3 Report
This is a very interesting study, considering the unique group of liver transplant patients.
My small remarks/comments are below:
1. A diagram of how to conduct the study would be useful
2. Not all abbreviations are explained correctly, e.g., WC, the explanation appears on line 138, and the abbreviation is used for the first time on line 129 - you should correct it and read the entire text from this angle.
3. Statistical analyses were selected correctly and presented properly,
4. Results
Table 1 lacks the units of the studied parameters; it is not enough to write the medians, e.g., in the case of lipid metabolism - this should be completed,
Moreover, converting this table into PDF resulted in "cutting off" information about overweight - this should be corrected. Most tables are missing units.
5. The discussion is well-conducted, but the conclusions are too extensive, and the authors' findings and opinions can be lost, so I suggest adding conclusions in a more condensed form at the end.
6. I have a question how the authors explain such a change in eGFR before and after OLT, the age of +10 years alone does not fully explain it. 7. What are the criteria for metabolic syndrome? I didn't notice this information in the article.
Minor editing of English language required.
Author Response
This is a very interesting study, considering the unique group of liver transplant patients.
My small remarks/comments are below:
1. A diagram of how to conduct the study would be useful
R. We have modified the manuscript and added the study flow-chart.
2. Not all abbreviations are explained correctly, e.g., WC, the explanation appears on line 138, and the abbreviation is used for the first time on line 129 - you should correct it and read the entire text from this angle.
R. We have modified accordingly.
3. Statistical analyses were selected correctly and presented properly,
R. We thank the reviewer.
4. Results
Table 1 lacks the units of the studied parameters; it is not enough to write the medians, e.g., in the case of lipid metabolism - this should be completed,
Moreover, converting this table into PDF resulted in "cutting off" information about overweight - this should be corrected. Most tables are missing units.
R. We have worked on the word program and have modified the table extensively.
5. The discussion is well-conducted, but the conclusions are too extensive, and the authors' findings and opinions can be lost, so I suggest adding conclusions in a more condensed form at the end.
R. We have revised the conclusion according to reviewer comments.
- I have a question how the authors explain such a change in eGFR before and after OLT, the age of +10 years alone does not fully explain it.
R. We extend our sincere appreciation to the reviewer for their valuable feedback. We have thoroughly discussed on the rapid decline in eGFR observed in our study. In our modest opinion, we believe that various factors might have contributed to this decline, including worsening of metabolic parameters, changes in blood pressure, diabetes onset, and potential iatrogenic glomerular damage due to immunosuppression and other medications. Additionally, we acknowledge the impact of aging on kidney function over the years of follow-up. - What are the criteria for metabolic syndrome? I didn't notice this information in the article.
R. We initially just reported the reference. We have now reported the definition of Metabolic Syndrome in the text.